# Historical-Cultural Sustainability Model for Archaeological Sites in Mexico Using Virtual Technologies

**Angel Geovanni Ambrosio Arias** [1], **Jesús Jaime Moreno Escobar** [2,*] , **Ricardo Tejeida Padilla** [1] and **Oswaldo Morales Matamoros** [2]

1   Escuela Superior de Turismo, Grupo de Investigación en Sistémica y Turismo, Instituto Politécnico Nacional, 07630 Ciudad de México, Mexico; aambrosioa1100@alumno.ipn.mx (A.G.A.A.); rtejeidap@ipn.mx (R.T.P.)
2   Escuela Superior de Ingeniería Mecánica y Eléctrica, Grupo de Investigación en Sistémica y Turismo, Instituto Politécnico Nacional, 07340 Ciudad de México, Mexico; omoralesm@ipn.mx
*   Correspondence: jemoreno@esimez.mx; Tel.: +52-55-5729-6000 (ext. 54639)

**Abstract:** The use of virtual and immersion technologies has expanded considerably due to their impact on user experience, economy, knowledge, and sustainable conservation of cultural heritage according to studies conducted in various parts of the world in different disciplines (architecture, economy, entertainment, health, tourism, etc.), including on tourism in Mexico. These technologies are used in some archaeological sites, but development and implementation are scarce due to the lack of economic strategies, infrastructure, and human capital, which are preventing the sustainable exploitation of those sites, although some of these sites have met the basic requirements for providing a better experience to visitors. However, these sites should be studied to propose integral solutions not only to improve the tourist experience, but also to assist in their protection, conservation, and sustainable development. Here, we used knowledge from the soft systems methodology and the hologram generation system to generate proposals to solve the problem described. The result is a sustainable historical-cultural model based on the systemic approach, whose objective is to positively impact the visitor experience while maintaining harmony with the environment.

**Keywords:** virtual and immersion technologies; soft systems methodology; hologram generation system; cultural tourism; historical-cultural diffusion; sustainability

## 1. Introduction

The desire to transform our physical environment and reality has continually evolved; it was not until the 20th century that it became feasible to quickly create, store, and retrieve information. The Industrial Age saw rapid advances in the ability to build, deconstruct, and modify physical structures with relative ease [1]. However, each change in reality was somewhat permanent, given the persisting ability to dismantle structures, so changing the information of something existing required the creation of a physical artifact or a modification to represent that change.

With the arrival of computers in the Information Age, all information could be presented digitally; large amounts of data occupying a small space could now to be stored, manipulated, and recovered quickly. Due to this ability to instantly modify and retrieve information, a better method of changing our environment arose [1], leading to a renewal in the concept of transformation. The concept of virtualization emerged. According to Brodkin [2], virtualization refers to the creation of a virtual version of some resource through software. Highlighting the previous concept, a wide range of tools have been introduced to carry materials, including natural and human resources, etc., toward

virtualization through different technologies such as augmented reality (AR), virtual reality (VR), and holography.

In Mexico, virtual technologies are already used in different areas of daily life, including architecture, sports, medicine, the arts, tourism, industry, entertainment (television, film and video games), and education. Within tourism, these technologies have been used to create simulations of museums, archaeological sites, historical sites, and other points of interest. In addition, they allow viewing maps for sharing information [3].

The National Institute of Anthropology and History (INAH, Instituto Nacional de Antropología e Historia) launched a program called INAH Places to allow nationals and foreigners to take virtual tours using images and text that provide information only, but do not consolidate the use of virtual and immersive technologies since they do not improve the user experience or support sustainability. By implementing an application used with simulation technologies, the archaeological site of Xochicalco and the site museum visitors in the State of Morelos can view 10 pieces in 3D in the exact place where they were found in that heritage site to familiarize themselves with these artifacts and their context, improving understanding of the importance of this place, which was declared a World Heritage Site by UNESCO in 1999. The Xochicalco AR México, for mobile electronic holographic devices such as cell phones and tablets, allows the public to identify places where they can activate the augmented reality viewer through the application's icon, which is located at 10 points within the site [4].

At the archaeological sites in Mexico, financing for the development of technologies is insufficient and technological instruments for promotion and commercialization are lacking, so a plan has been implemented for the preservation and restoration of the existing cultural heritage, such as historical buildings, including taking advantage of sustainable and available resources [5]. The number of visitors to archaeological sites is low; these sites are sometimes little known due to the deficiencies mentioned above. The scarcity of resources impacts the environment of the archaeological site; the lack of preparation and training of some tourist guides directly affects the tourist experience. For example, at the El Tepozteco archaeological site, the lack of sustainable heritage conservation and the high number of visitors are reflected in the damage caused to the cultural and natural heritage in the area, which impact the tourist experience, as it makes the site unattractive.

Based on the aforementioned considerations, we focused on verifying which components form a sustainable historical-cultural model, helping to fill the information gap and support the comprehensive development of the environment. By analyzing the different archaeological sites in Mexico, we selected El Tepozteco as a study reference because this archaeological site has been negatively affected by natural and human causes. Hence, we aimed to construct a model to allow the user to use virtual and immersive technologies to provide quality information about the sit to the visitor, seeking to raise awareness about the conservation of cultural heritage and protected natural areas. The archaeological site, being part of the local heritage, is subject to the perception of the local population about the evolution of nature-based tourism; the consequent impacts on their well-being are crucial to promoting ecotourism and achieving sustainable development [6].

The priority of heritage management is the conservation and restoration of archaeological sites because the dissemination of their historical and cultural wealth to society is sometimes lost. This has been the case of the Kukulkán Pyramid in Chichén Itzá, another archaeological site located in Yucatan, which was forced to close its doors to the public as a strategy for its conservation due to the deterioration caused by the visitors.

Considering the aforementioned challenges faced by other archaeological sites in Mexico, the competition for this type of tourist attractions in order to attract tourists is increasing [7]. Comparisons in the use of VITs are required to strengthen the competition  within El Tepozteco archaeological site to prioritize the management of this tourist attraction and to responsibly increase the tourists demand for the archaeological site. The increase of the tourist demand  can generate income that supports restoration and conservation for the continuous study of the area.

Due to the advances in the field of virtual and immersion technologies and a meticulous work process, some archaeological sites that have been over-exploited have been made accessible again through INAH Places program in a non-intrusive way. This creates an innovative experience for the visitor, generating a tourist advantage compared to archaeological sites that do not use these new technologies. In Mexico, there is no tool capable of highlighting archaeological tourist attractions that enriches the experience of visitors to this cultural heritage.

Currently, the Tourism Wellbeing program is being implemented by the Mexican government. One of the sections of this program mentions four proposals for developing sustainable and modern tourism:

- Optimize the use of natural resources in harmony with tourist activities, always seeking the preservation of natural capital, by incorporating new technologies and promoting a culture of environmental conservation.
- Strengthen the capacity of the tourism sector in Mexico to innovate and address mega trends and technological changes.
- Consolidate the link between scientific, technological, and innovation activities with the national tourism industry.
- Promote smart tourist destinations and facilitate the flow of people [5].

Compared to the use of mobile AR technology, the historical-cultural diffusion model with virtual and immersive technologies is based on the use of holography, allowing all visitors access without a mobile device, making the holographic pyramid more accessible and efficient to improve people's experience at the archaeological site.

In Mexico, both the AR program, developed by the government and managed at the Xochicalco archaeological site, and the system of auditory documentation, linked by QR code, contribute to the conservation and dissemination of the cultural heritage, positively impacting the visitor's experience.

As in Xochicalco, and especially during the COVID-19 pandemic, we must rely on the enhancement of tourist destinations and different technological tools to attract new markets and maintain the interest of those already reached to survive a growing wave of tourist competition. An important aspect to highlight is that young people who are most skilled in the use of technologies, called "Digital Natives", have the desire to use virtual and immersion technologies for archaeological sites. Their attraction to historical and cultural heritage has increased, having a greater acceptance of the inclusion of these technologies in tourist attractions.

The archaeological site of El Tepozteco is located in the municipality of Tepoztlán in the state of Morelos, a city founded by the Xochimilcas-Tepoztecas (although various authors mention the Aztecs as founders). El Tepozteco is a sanctuary of the lordship of Tepoztlán an architectural complex composed of a temple, its dependencies, a square, and a residential area. Since this site has suffered considerable wear and tear due to visitors and natural phenomena, we selected this location for this project to increase its value as a tourist attraction and support the need for cultural heritage and meet the demands of the population.

Our main objective was to diagnose the current situation of the cultural and historical heritage at the archaeological site of El Tepozteco to design a sustainable historical-cultural diffusion model using virtual and immersive technologies (VITs) by applying the systems thinking paradigm.

To reach our goal, our secondary objectives were as follows: (1) to identify the components, actors, and relationships of the system under study; (2) to study the advantages and disadvantages of using VITs for the object under study and its context; (3) to define the root of the relevant systems to integrate a sustainable construct; and (4) to contrast the conceptual model with reality based on the construction of a prototype with VITs.

Therefore, our hypothesis was as follows: VITs in the diffusion of history and culture improve the tourist experience and contribute to the preservation of heritage. Hence, to achieve our main

objective and to test our hypothesis, the systemic method and its treatise were used through a systemic methodology, applying a complementary procedure for the integration of VITs.

## 2. Literature Review

### 2.1. Cultural Heritage, Cultural Tourism, and the Tourist Experience

Heritage is the legacy received from the past, lived in the present, and passed on to future generations. The cultural and natural heritage is an irreplaceable source of life and inspiration, based on the convention about protection of the world, and cultural and natural heritage.

Here, the following are considered cultural heritage: (1) monuments: monumental architectural, sculpture or painting works, elements or structures of an archaeological nature, inscriptions, caverns, and groups of elements that have exceptional universal value from the point of view of history, art, or science; (2) groups: groups of constructions, isolated or assembled, whose architecture, unity, and integration in the landscape give them an exceptional universal value from the point of view of history, art or science; and (3) places: works of humankind or joint works of humans and nature as well as areas, including archaeological sites, with exceptional universal value from the historical, aesthetic, ethnological, or anthropological points of view [8].

Within cultural heritage, some objects cannot be viewed by visitors due to lack of space or their restoration. Sometimes, despite these objects being accessible, they can only be seen from a specific site since they are fragile or they simply cannot be transported [9]. The cultural heritage of humanity has a clear intangible dimension, as it is seen to result from "long developments and the traditional transfer of knowledge in particular societies", as well as through influences and cross-fertilization among different cultures and civilizations [10].

World Heritage sites are economically advantageous. They create jobs, promote local activity through crafts, promote tourism, and generate income. From this perspective and in addition to heritage conservation, the Convention on the Protection of World, Cultural, and Natural Heritage is a source of socioeconomic development [11]. In the 21st century, there is a trend towards the use of cultural heritage in the construction of competitive brands and adding value because cultural heritage can function as a differentiation tool never equaled by competition. Cultural heritage research could aspire to delivering higher quality results. The research results should be used to help public policy agencies willing to reform, e.g., by demonstrating the social and economic impacts of the sector and measuring change.

The cultural heritage brand is a brand with positioning and value proposition based on cultural heritage. Although the term "cultural heritage brand" is a relatively recent introduction, the concept of "cultural heritage and heritage brand" is increasing in popularity [12]. Cultural tourism appeals to the memory of humankind and its creation; it is presented as an alternative or complement to the typical tourism of the sun and beach, and it may also help to economically reactivate certain cities or regions [13].

Understanding the patrimonial assets of a country is an important axis for tourism, allowing retaking the identity that has been lost or dissipated due to globalization. This type of cultural trip, allows people to come into contact with several cultures and brings the person closer to their identities, has experienced a boom in recent years. According to Ibañez and Rodríguez [14], there are no current references in the literature that clearly establish whether cultural tourism is becoming a massive activity; thus, cultural trips as part of alternative tourism are developed around sustainability. This kind of tourism can be considered within alternative tourism, as long as it allows contact between the culture and traditions of the receiving community and the visitor, respecting their integrity, taking care of their natural environment, and granting fair and equitable benefits.

According to the above, cultural tourism is largely influenced by cultural heritage and archaeological sites. However, the conservation of heritage and the environment should not be forgotten, since pro-environmental behaviors (PEBs) represent behaviors that cause minimal harm to

or even benefit the environment [15]. This vision can be applied to the employees of archaeological sites. Throughout the history of tourism, tourist experiences obtained when practicing PEBs were initially defined by focusing on differentiating the activities performed within tourism from those of daily life.

Taking up what was said by Cohen [16] "Tourism is essentially a temporary reversal of daily activities, it is a situation without work, without care, without saving" referred to the search for strangeness and novelty as key elements. Likewise, a person in search of a tourist experience may temporarily relax and visit a place away from home to experience a change [17]. The differentiation between everyday life and the tourist experience was also highlighted by Turner and Ash [18], suggesting that the temporary distance of tourists from their normal environment allows them to suspend the power of norms and values that govern their daily lives and think about their own lives and societies from a different perspective.

Starting the 1990s, the differentiation between the tourist experience and daily life lost strength. Lash and Urry [19] defined the distinctions between daily life and tourist experiences as "the end of tourism". Specifically, they argued that experiences once limited to tourism, including the pleasure of viewing distant places and the pleasure of participating in aspects of other cultures, are currently accessible in various contexts of everyday life. In the media age, for instance, various tourist sites and attractions can be enjoyed through video and virtual technologies and immersion within the comforts of home. Similarly, the proliferation of simulated environments could gather multiple sites and views from around the world in museums, theme parks, or shopping malls [20]. Although the tourist experience is a way to escape everyday life, the experience is composed of a range of different motivations that allow tourists to satisfy different needs that they cannot satisfy in their daily lives.

*2.2. Historical-Cultural Dissemination and the Theory of Communicative Action*

Dissemination is the elaboration of messages accessible to the entire public, regardless of whether they deal with topics of common interest. For the correct dissemination of history and culture, a series of approaches must be followed to be considered as comprehensive dissemination, among which the following stand out:

- Dissemination is not only the sending of content from one sector to another but is the initial feed of information that is digested and processed to make it understandable to potential communities.
- Scientific dissemination must be understood as a communication and social need.
- Dissemination has to exceed the mere transmission of information, being understood as comprehensive communication.
- In the progressive sense, dissemination must be understood as a process that contributes to developing policies to benefit heritage.
- Dissemination must be handled democratically; no interests should be placed before the information disseminated or in the media or channels where it is disseminated.

Cultural and historical dissemination consists of planning, organizing, and performing activities to publicize the expressions of culture, through amateurs, professionals, experimental groups, or specialized groups, from bodies created for this purpose.

Habermas [21] suggested the need to achieve universal pragmatics, i.e., a science on linguistics capable of integrating universal and validated language science based on universalized structures that is valid in any communicative context. In this new science, there are the conditions that make communicative reason possible. These conditions are determined by the modalities of action realized by the subjects. Therefore, Habermas resorted to the review of sociology from Durkheim and formulated the types of social action:

1. Strategic or teleological action: associated with conscious purpose.

2.  Action regulated by norms: associated with shared values and legitimized by the subjects in social life.
3.  Dramaturgical action: associated with the full manifestation of individual subjectivity.
4.  Communicative action: the interaction between two subjects capable of communicating linguistically and of performing actions to establish an interpersonal relationship.

Likewise, Habermas conceived the possibilities of achieving understanding: "The concept of understanding refers to a rationally motivated agreement reached between the participants, measured by validity claims susceptible to criticism. Validity claims (prepositional truth, normative rectitude and expressive truthfulness) characterize several categories of knowledge embodied in symbolic manifestations or broadcasts" [21].

Communicative action is a part of social action, which makes it a determining factor in the socialization process. Currently, this is essential for understanding the relevance of the mass media in the formation of world images of subjects. The communicative dynamics define cultural reception and reproduction, social integration, and the development of personality and personal identity.

Habermas, in Volume II Critique of Functionalist Reason [22], criticizes Talcott Parsons's Theory of Society given the problem that exists with their normative conception of action in systemic terms:

This gives rise to the false impression that the functional analysis of action complexes refer to the conception of society as a self-regulating system. However, if the concept of the world of life is introduced as a complement to that of communicative action and the world of life is understood as a contextualizing background for the processes of understanding, then the reproduction of the world of life can already be analyzed from different functional points of view.

This is fixed regarding the idea of cultural determinants in the orientation of action of the subjects.

*2.3. Holography*

Holography is a branch of modern optics that has considerably impacted science and technology. It can be understood as a photographic technique that allows obtaining a three-dimensional image through the use of a projection. This technology was created by the Nobel Prize winner in Physics, Dennis Gabor, in 1947, in a quest to improve the resolution of the electron microscope [23]. Its use has expanded to various areas including arts and security; the latter can be exemplified in banknotes that are used in everyday life, since a hologram is difficult to counterfeit due to the technology used in its creation. Within art, holography is used within museums to conserve the exhibited cultural heritage. A recent study by Caggianese et al. [24] showed how holography can be applied in museums. Within this project, the used Interactive projection system was based on the ghost effect by Pepper, which creates a 3D hologram illusion by reflecting an image using a glass pyramid placed at a 45-degree angle from the public, as shown in Figure 1.

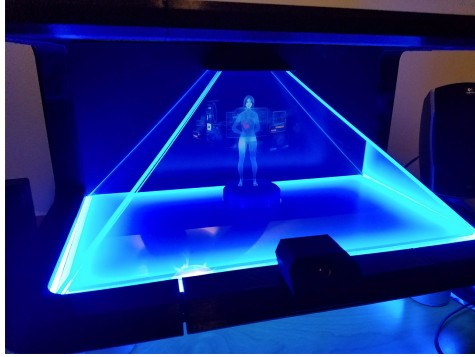

**Figure 1.** View of a hologram inside the holographic pyramid.

The ability of holography to transmit real-appearing images could better preserve cultural heritage, since the presence of the real object is not required, plus it allows interaction with the object, improving the user experience.

### 2.4. VITs in Similar Studies

VITs allow enhancement of cultural heritage for visitors before, during, and after the visit. Businesses can benefit from increased spending, return intent, and positive word of mouth, whereas visitors receive a personalized, educational, memorable, and interactive experience.[25]. By allowing the user to see the real world, with virtual objects overlapping or composite with the real world, this medium complements the environment, improving the user experience and generating added value. Augmented reality has been increasingly implemented to enhance visitor experiences, and tourism research has long understood the importance of creating memorable experiences [26]. Such is the case of the Xochicalco AR mobile application used in the Xochicalco archaeological zone.

In the context of tourism, VITs have been proposed as tools to improve experiences [27], increase tourism accessibility, and contribute to heritage conservation [28]. Empirical studies associated VR with greater attention, interest, desire, and action of tourists towards destinations, as well as high enjoyment of the attraction, resulting in a preference towards a destination [29]. Barrado and Timón [30], in their work *The Historic City, Its Transmission and Perception via Augmented Reality and Virtual Reality and the Use of the Past as a Resource for the Present: A New Era for Urban Cultural Heritage and Tourism?*, proposed the usage of tools based on VITs to generate positive and negative effects in the process of interpreting urban cultural heritage. They claimed that these technological tools facilitate the dissemination of information, providing a quick, easy, and innovative approach to knowledge that is also highly valued by the user. In addition, they affirmed that the use of these technologies within the promotion and marketing of tourist sites makes them valuable tools that provide the visitor with a better experience, as well as help attract new tourist profiles. Lastly, in this work, the authors discussed the perception and transmission of cultural heritage, so the function of VITs as mechanisms for protection and conservation can be emphasized through the digital recovery or recreation of cultural assets.

Despite the poor development of these technologies in the tourist environment, a model that impacts the user experience with augmented reality applications was recently constructed. In Han and Dieck [31], the acceptance of this type of technology depends on three factors: (1) perceived attributes of innovation, (2) visitor benefits, and (3) resistance from the visitor, demonstrating a holistic approach to the situation in which people want to adapt VITs in the dissemination of historical and cultural heritage is necessary.

Using new technologies is difficult because it requires a long learning period and is expensive, which slows absorption. However, this applies to all innovative technologies, and studies of technological evolution suggest the eventual appearance of a phase of commodification in which cost decreases, absorption increases, and technology evolves rapidly [32].

## 3. Methodology

The systems thinking, or systems science, approach is a scientific method used in the understanding of human behavior that is based on three pillars: general systems theory, cybernetics, and the theory of communicative action, to understand vision, a fundamental transformation of our way of thinking, of our way of perceiving, and of our way of valuing. This holistic approach is essential for adopting a systemic paradigm to understand the nature of all our realities.

Systems thinking is indispensable when dealing with dynamic entities or systems that do not integrate with homogeneous elements; therefore, it cannot be applied to laws constituting our current mathematics without denaturing them. The additive law of elements (the commutative, the associative, and the distributive) [33] is why, based on Checkland's soft systems methodology (SSM) [34], each of the seven stages (Table 1) was developed.

Tourism has become a dynamic entity requiring complex methodologies, and systems science provides new elements and tools for theoretical and praxiological approaches to tourism. When looking for advantages and disadvantages of different methodologies, the SSM could be used. According to their complementarity, the treaty of the systemic method with the structuring of the use of VITs generates a breakdown in a procedure to reach the main goal and test the hypothesis. Within the SSM framework, we planned to use a holography prototype for comparison with the conceptual model using the hologram generation system [35] as an alternative methodology.

**Table 1.** Soft systems methodology.

| Stage | Description |
| --- | --- |
| 1. Unstructured Situation | The current problematic situation is investigated, including the actors involved and their vision. |
| 2. Structured Situation | A diagram is constructed with information collected from the situation showing the processes and relationships with an observer's point of view. |
| 3. Identification of Relevant Systems | Tentative construction of a definition to improve the problem situation, generating a root definition for each system. |
| 4. Integration of Conceptual Models | From the definition, a conceptual model is constructed representing the activities necessary to the system performance. |
| 5. Comparison of the Conceptual Model | The constructed conceptual model is compared with what exists in the structured-problem situation. |
| 6. Feasible and Desirable Changes | Possible changes are defined as desirable and feasible. |
| 7. Implementation of Changes | Actions are defined to make the changes suggested in the previous stage. |

The procedure to achieve our objective and test our hypothesis is reflected in the development of the seven-stage methodology as follows:

Stage 1: In this first stage, the unstructured problem is diagnosed and projected, so the experience of the researcher is essential, requiring a personal perspective on the situation and environment. Reflecting on the previous diagnosis, the main actors are the archaeological zone along with the personnel who work there, information technologies, and visitors. Within the environment, the National Institute of Anthropology and History, the Ministry of Tourism, and the Ministry of Communications and Transportation can be highlighted. The interpretations were performed in conjunction with visitors and workers in the archaeological zone through their experience, allowing the identification of advantages and disadvantages.

Stage 2: This stage details an enriched vision of the situation environment where the problem and the actor interrelationships within the environment are reflected, showing stable relationships, conflicts, and non-existent relationships.

Stage 3: In this stage, the root definitions are generated, and the CATWOE mnemonic (Table 2) proposed by Checkland is used as a list to verify the actor roles.

**Table 2.** CATWOE mnemonic.

| | | |
|---|---|---|
| **C** | Client | Visitors to the archaeological zones and tourists |
| **A** | Actors | Developers, tourist guides, town population, government, and local authorities |
| **T** | Transformation | Transformation process and its feedback |
| **W** | Weltanschauung | Worldview obtained from the observation of the environment and from a series of interviews collected on visits to the destination, as well as from research in various media in the State of Morelos and national sources. |
| **O** | Owner | Population of Tepoztlán (Magical Towns Committee) |
| **E** | Environment | Formed by the local government, Secretary of Tourism, National Institute of Anthropology and History and Laws, and regulations for the use of technological tools |

The proposed root definition for this stage was: the construction of a sustainable historical-cultural diffusion model to enrich the tourist experience and contribute to the conservation of the El Tepozteco archaeological site using VITs.

Stage 4: In this stage, the conceptual model was constructed based on the results of the previous stages, including the interpretation of the interrelationships.

Stage 5: The conceptual model and reality were compared with the help of an alternate prototype to the holographic pyramid, an online survey, an interview with the manager of Xochicalco in which virtual and immersion technology is already used (AR), and other studies considering innovative technologies.

Stage 6: In this stage, desired and feasible changes were identified, emerging from the contrast where the differences between the prototype alternative to the holographic pyramid and reality were recognized.

Stage 7: In this stage, the actions to improve the situation emerged from the experts, and owners evaluation and approval were verified, to guarantee the sustainability of El Tepozteco in the future.

### 3.1. Alternate Prototype to the Holographic Pyramid

One system for observing holograms is the four-sided holographic system, or the holographic pyramid. It is a pyramidal display with four projections showing objects in a single 3D image. The developed holographic device uses the holography reflection principle for the reproduction of three-dimensional images. This phenomenon occurs when light rays collide with a surface, deviate, and return to the medium from which they were produced, forming an angle equal to the incident light. The proposed prototype creates holographic three-dimensional images, giving a volume to certain objects, thus facilitating their study and manipulation. The hologram generation system consists of three phases: capture, coding, and representation.

### 3.1.1. Capture Phase

The capture phase is composed of a set of images captured from the El Tepozteco archaeological site from various points in it. This phase is subdivided into two subphases:

1.  The capture of images was made with a Sony CyberShot Camera Model DSC-HX200V in RAW format, trying to avoid contamination of the capture by people (Figure 2). Sony CyberShot DSC-HX200V has the following features:

    -   Depth: 3.7 in.
    -   Sensor Resolution: 18.2 Megapixel.
    -   Digital Video Format: AVCHD and H.264.
    -   Optical Sensor: TypeExmor R CMOS.
    -   Effective Sensor Resolution: 18.2 Megapixels.
    -   Total Pixels: 18.9 Megapixels
    -   Image Recording Format: JPEG and MPF.
    -   Light Sensitivity: ISO 100-12800, ISO auto.
    -   Exposure Metering: center-weighted, multi-segment, spot.

2.  The lighting was natural, since the dimensions cannot be used with lighting equipment, taking care to capture at dusk, when the sky was cloudy.

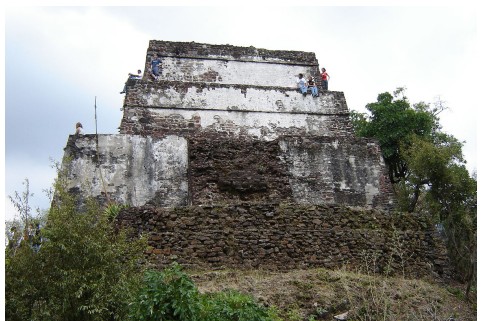

**Figure 2.** El Tepozteco.

### 3.1.2. Coding Phase

The image coding or processing phase is a subsystem complementing the general operation of the prototype. This subsystem works with toolboxes from MATLAB, detecting the green pixels closest to the Chroma Key (in this case, a white background was used, since it is not possible to use a green screen due to the dimensions of the objective), and thus replace them with black pixels. The general coding algorithm is shown below:

1.  Reading the image with original background.
2.  Create an .avi video extension file, assigning it a name.
3.  The video created to integrate the images starts.
4.  A variable is created to save the image as $I$.
5.  The obtained threshold value $\mu$ is assigned.
6.  The RGB color space is transformed to YCbCr opponent color space.
7.  The white background $I_{bg}$ is removed (Figure 3) and replaced by the image with a black background, obtaining the $I_c$ image.
8.  The $I_c$ image is converted to the RGB color space as $\widehat{I_i}$.
9.  An $\widehat{I_i}$ image is added to the corresponding frame.

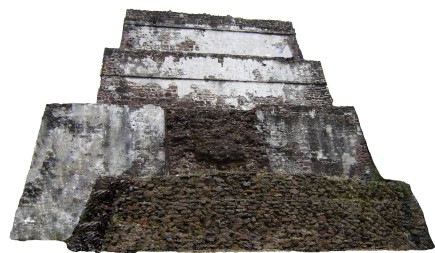

**Figure 3.** Edited image of El Tepozteco before the black background.

### 3.1.3. Representation Phase

The representation phase consists of two elements for the correct display of the hologram. The first element is a projection device, which projects a video of the target with a black background in the holographic pyramid. The second one is an alternative to the holographic pyramid consisting of a three-sided pyramid, allowing viewing the object in three-dimensional form:

1.  Projection Device: To determine the type of pyramid to be used, it is necessary to consider the hologram projection method. Due to the size of the object to be projected, a projection must allow an acceptable scale of the object.
2.  Alternate Proposal to the Holographic Pyramid: After determining the form of projection, the alternate proposal to the holographic pyramid to be used is constructed. Since the projection corresponds to a large object, with the face of the pyramid refracting the projected video at 45 degrees, it does not lose image quality when scaling.

The holographic pyramid is a device used for various means such as advertising, information media, marketing, etc., allowing people to project elements in three dimensions (people, objects, products, or almost any object). This reproduction captures the attention of users.

For the alternate prototype (Figure 4), unlike the holographic pyramid, it consists of three sides composed of a transparent material, so people can see what happens in the center of the pyramid from all directions, which is the function of holography. This function is generated by the creation of the three reflections.

One technical consideration for the construction of the proposal is the angle at which the viewer observes the projection. Therefore, we propose using a right-viewing angle to observe the hologram comfortably, displayed in the center of the holographic device. In addition, the dimensions and quality of the projection holographic device were considered.

To calculate the dimensions in the alternate design of the holographic pyramid, the maximum height was estimated for a 45-degree pyramid. In a pyramid, the maximum height is equal to the distance from the tip of the pyramid to the center of its base, from which only one face is taken.

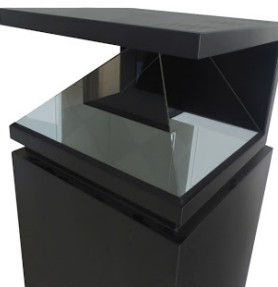

**Figure 4.** Alternate prototype of the holographic pyramid.

### 3.2. Survey

To collect information for the conceptual test, a questionnaire was designed to verify the knowledge and experience of people about VITs. In this case, we created a questionnaire using Google Forms given the social distancing requirements in Mexico because of the pandemic caused by the SARS-CoV-2 virus (COVID-19). The questionnaire was disseminated via social networks such as Facebook, Twitter, and WhatsApp to the general population, and we received 202 responses. Two out of these 202 questionnaires were rejected because they were answered as a pilot test. Hence, 200 questionnaires were used for analyses, which asked questions about eight aspects relevant to this research:

1. Participant's general information
2. Knowledge of virtual technologies
3. Most common type of virtual technology/immersion used
4. Educational experience
5. Entertainment
6. General experience
7. Aesthetics and accessibility
8. Sustainability.

### 3.3. Expert Interview

An interview was conducted as a research tool with an expert on tourism, archaeological sites, and VITs to support the current research on VITs and their impact on archaeological sites, based on their experience.

## 4. Results

As briefly described in the Methodology section, the study and interpretation of the current situation of the historical and cultural heritage of El Tepozteco resulted in recognizing the interrelationships (Figure 5 and Table 3) among the distinct actors.

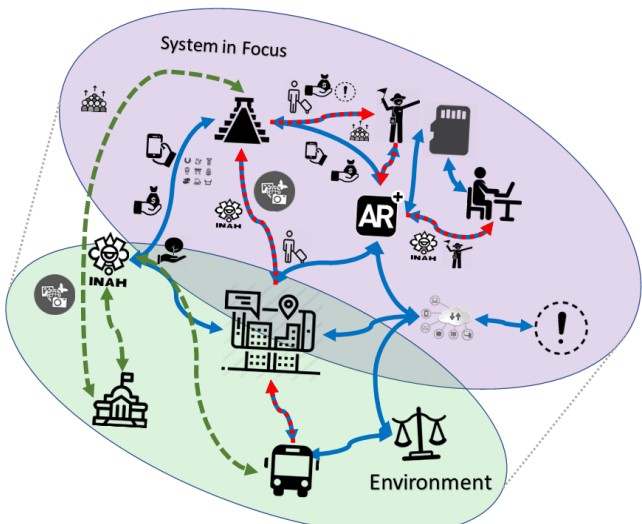

**Figure 5.** Enriched vision.

**Table 3.** Enriched vision relationships,

| Actors | Relationship |
|---|---|
| Archaeological Site, Tourists, Visitors Guide | The tourist guides lack training, which distances visitors from their services, and then affecting the influx to the archaeological site because of the lack of information. |
| Ministry of Communications and Transportation (SCT, Secretaría de Comunicaciones y Transportes), INAH, Government | A relationship among these elements does not exist, since the government does not have the power to license the use of new technologies for taking images, preventing INAH from taking pertinent actions. |
| Archaeological Site, INAH, Local Government | Non-existent relationship due to the lack of support from the local government of INAH for the conservation of the archaeological site. |
| Tourists, Developers, Virtual Technologies Guide | Relationship in conflict, since tourist guides are resistant to change and the use of virtual technologies, causing developers to reject and share information about the place. |
| Virtual Technologies, INAH, Developers | Relationship is non-existent since there is no interest on the part of INAH to use new virtual technologies as a method of conservation and information. |
| Local Government, Local Population, Archaeological Site, Ministry of Tourism (SECTUR, Secretaría de Turismo) | Relationship in conflict, since the interests of the local government and the population are contrary to the conservation of the archaeological site, due to corruption. |
| Visitors, Archaeological Site, INAH | Relationship in conflict; the archaeological site does not offer enough to generate a good experience for visitors, and visitors do not abide by INAH recommendations. |

*4.1. Conceptual Model*

The conceptual model was integrated as a result of the actor interrelationships analysis within the environment, showing stable, conflicting, and nonexistent relationships (Figure 5).

During the model integration, the minimum functions were established to achieve the system purpose, and we proposed the following systems:

1. A planning system to carry out collection activities and review the data to be used, as well as to define objectives to allow smooth operation of the dissemination and conservation.
2. A development system to develop prototypes, content, and applications with virtual technologies, presenting these applications to the owner for validation.
3. An implementation system to apply virtual technologies within the archaeological site to verify their correct function and/or repair errors, meeting the established objectives.
4. A control system to assess the operation and fulfillment of the objectives, implementing strategies and providing feedback to previous systems.

The above four systems directly impact the visitor experience and the archaeological site conservation. The proposed model (Figure 6) acts as a prototype due to the evolution of technologies. Likewise, the advances in the methodology allow modifications to enrich the model, since Checkland's methodology provides flexibility in the stages, depending on what is required during the investigation.

*4.2. Online Survey*

4.2.1. Overview and Knowledge of VITs

With respect to the questionnaire respondents, they ranged in age from 16 to 76 years, with an average of 34 years. Hence, adolescents and young adults predominate, representing 64% out of the questionnaires returned. Their education level is shown in Figure 7.

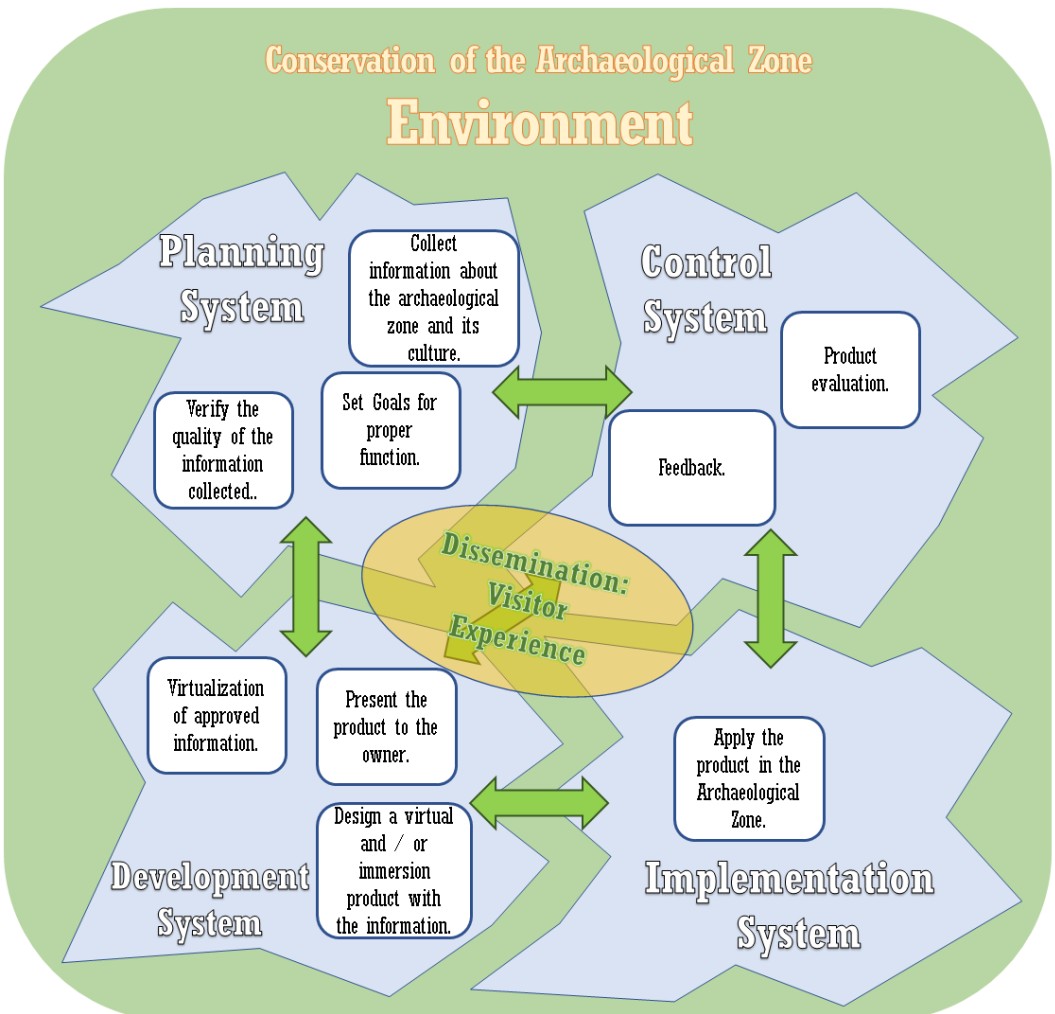

**Figure 6.** Conceptual model.

## Scholar Level

200 Answers

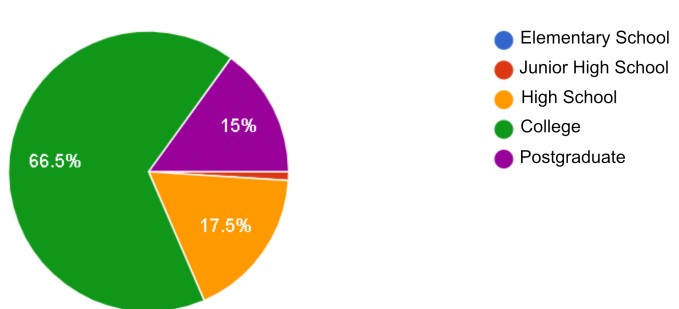

**Figure 7.** Educational level of questionnaire respondents.

Five educational levels (elementary, junior high school, high school, college, and postgraduate) were considered as a basis to identify the attraction of different segments of the respondents to VITs, finding that 81.5% of total sample (66.5% with college level and 15% with postgraduate level

education) approved these new technologies, i.e., the higher the educational level, the higher the degree of attraction.

Regarding occupation, most of the users were professionals, reflecting the ability to use these technologies depending on their academic and economic preparation. The question: "Do you know what holography, augmented reality, and/or virtual reality are?" was used as a filter to separate people who did not know about the subject from those who did, to increase certainty when verifying the impact of these technologies on the relevant aspects previously listed (mainly from numbers 3 to 8). Of the total answers, 83.5% were affirmative (Figure 8), confirming that most people knew about at least one out of the three types of VITs highlighted in the filter question. Accordingly, the number of participants was reduced 167, since people who had no knowledge of the subject were rejected.

By analyzing the response to: "Have you used any of these holography, augmented reality (AR), and/or virtual reality holographic devices?", we found that the VIT most used by those surveyed who did know about VITs was AR, with 137 people, perhaps because this technology is easily accessible and most mobile phones and devices have the ability to use this technology, especially given the large number of video games and mobile applications.

AR was followed by VR with 112 participants. Although VR provides a more immersive experience, it is still not as accessible as AR since its use requires more developed holographic devices and specific places (video game centers, virtual travel experiences, museums, etc.).

According to the above, we found that holography was the least used technology of the three due to its lack of development in Mexico (regarding the holographic pyramid); only 31 out of the total participants knew about it. Although the filter was applied, 7 out of the 167 users did know about these technologies, despite not having used them, highlighting the importance of this topic for people. In the subsequent analysis, we emphasized discovering the impact of VITs on users. The results of this analysis are described in Figure 9.

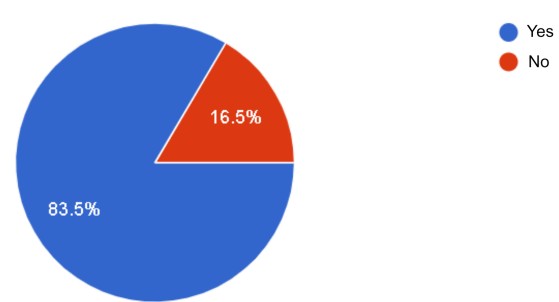

**Figure 8.** General knowledge about virtual and immersive technologies.

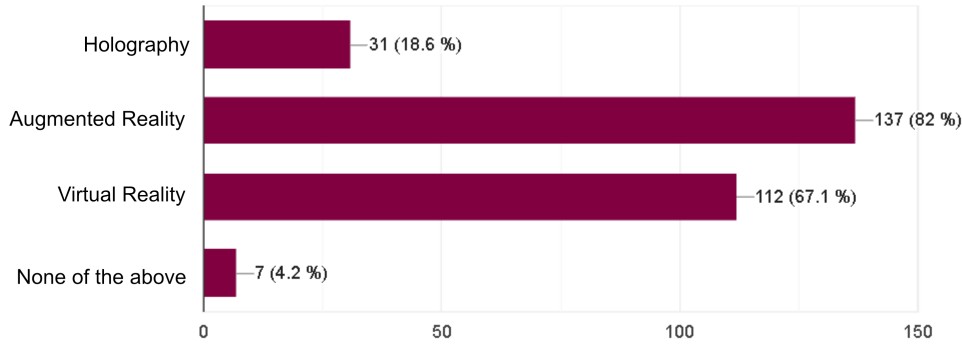

**Figure 9.** Usage of virtual and immersive technologies.

### 4.2.2. Education

Table 4 shows that the educational experience aspect is positively impacted by the VITs, allowing these technologies to be used as tools for the diffusion of new learning. In total, 70.6% of all the responses were positive, so the participants learned something new when using any of the VITs that were contemplated. In addition, 56.3% of people totally agreed that these holographic devices stimulate their curiosity to continue learning new things.

**Table 4.** Educational experience analysis.

| Question | Strongly Disagree | Disagree | Indifferent | Agree | Strongly Agree |
|---|---|---|---|---|---|
| I learned something new by using the holographic device. | 4 (2.4%) | 9 (5.4%) | 36 (21.6%) | 53 (31.7%) | 65 (38.9%) |
| The experience increased my knowledge. | 5 (3%) | 10 (6%) | 33 (19.8%) | 59 (35.3%) | 60 (35.9%) |
| It stimulated my curiosity to learn new things. | 2 (1.2%) | 8 (4.8%) | 18 (10.8%) | 45 (26.9%) | 95 (56.3%) |
| It provided me with a new learning experience. | 2 (1.2%) | 6 (3.6%) | 26 (15.6%) | 41 (24.6%) | 92 (55.1%) |

Most people saw VITs as a new learning experience, indicating the feasibility of the inclusion of new educational tools with this type of technology.

### 4.2.3. Entertainment

Table 5 depicts the impact of VITs on user entertainment. The vast majority of users (73.1% on average) reported a positive impact on fun because the aforementioned technologies improved the user entertainment experience. In addition, the content and use of these holographic devices are interesting to the public, so a more exhaustive investigation of how technologies attract people is needed.

**Table 5.** Analysis of entertainment.

| Question | Strongly Disagree | Disagree | Indifferent | Agree | Strongly Agree |
|---|---|---|---|---|---|
| Using the holographic device was fun. | 2 (1.2%) | 3 (1.8%) | 8 (4.8%) | 33 (19.8%) | 121 (72.5%) |
| Using the holographic device was entertaining. | 2 (1.2%) | 2 (1.2%) | 9 (5.4%) | 31 (18.6%) | 123 (73.7%) |
| Using the holographic device was interesting. | 0 (0%) | 4 (2.4%) | 7 (4.2%) | 34 (20.4%) | 122 (73.1%) |

### 4.2.4. Experience

Table 6 shows the analysis of general experience in the use of AR, VR, and holography, suggesting a positive contribution perceived by 59.9% of users; these technologies may help the visitor to enjoy their lived experience. An important result is the perception of people about using this type of holographic device for archaeological sites, museums, tourist sites, etc., with 73.7% of the participants fully agreeing that they should be used in the sites previously mentioned. This reflects the positive visitor experience produced with VITs. Despite these technologies depicting a virtualization of the environment or a digitization of some object, 86.3% of the participants wanted to visit or realistically experience the environment in a virtual way, providing a guideline for using these technologies as informative, dissemination, and promotional media.

**Table 6.** General experience analysis.

| Question | Strongly Disagree | Disagree | Indifferent | Agree | Strongly Agree |
|---|---|---|---|---|---|
| The holographic device positively contributedto my experience. | 3 (1.8%) | 4 (2.4%) | 16 (9.6%) | 44 (26.3%) | 100 (59.9%) |
| The holographic device helped me enjoy my visit more. | 2 (1.2%) | 9 (5.4%) | 12 (7.2%) | 46 (27.5%) | 98 (58.7%) |
| The holographic device gave me a significant experience on my visit. | 5 (3%) | 6 (3.6%) | 16 (9.6%) | 48 (28.7%) | 92 (55.1%) |
| The holographic device should be used in Archaeological Sites, Museums or various Tourist Sites. | 3 (1.8%) | 2 (1.2%) | 9 (5.4%) | 30 (18%) | 123 (73.7%) |
| After using the holographic device, I would like to visit the place that I walked through the holographic device. | 4 (2.4%) | 2 (1.2%) | 17 (10.2%) | 33 (19.8%) | 111 (66.5%) |

### 4.2.5. Aesthetics and Accessibility

Table 7 describes the users' opinions, indicating that the technologies are simple to use, since there are often trained personnel who support their management, and most mobile applications have an integrated tutorial or guide. In addition, people had a pleasant impression of the physical presentation of the holographic devices. An interesting aspect of these new technologies is the quality of the audio and video, which was enjoyed by the majority of the participants. Nonetheless, some 15 % of respondents were indifferent to this aspect.

**Table 7.** Analysis of aesthetics and accessibility.

| Question | Strongly Disagree | Disagree | Indifferent | Agree | Strongly Agree |
|---|---|---|---|---|---|
| Using the holographic device is easy. | 1 (0.6%) | 6 (3.6%) | 29 (17.4%) | 63 (37.7%) | 68 (40.7%) |
| The holographic device has a good presentation. | 2 (1.2%) | 1 (0.6%) | 28 (16.8%) | 55 (32.9%) | 81 (48.5%) |
| Audio and video quality are good. | 1 (0.6%) | 6 (3.6%) | 25 (15%) | 59 (35.3%) | 76 (45.5%) |

### 4.2.6. Sustainability

Finally, we determined the respondents' opinions about the usage of these technologies as a method to improve sustainability of archaeological sites. although they were not experts on the subject, the responses allowed us to visualize the impacts of using these technologies as tools for sustainable historical-cultural dissemination in these areas. As shown in Table 8, few people disagreed with the statement that holographic devices damage the environment (3.6%); thus, it could be argued that the holographic devices are kind to the environment.

**Table 8.** Analysis of sustainability.

| Question | Strongly Disagree | Disagree | Indifferent | Agree | Strongly Agree |
|---|---|---|---|---|---|
| Does the holographic device damage the image of the environment where it is used? | 4 (2.4%) | 2 (1.2%) | 21 (12.6%) | 47 (28.1%) | 93 (55.7%) |
| Do you believe that holographic devices could help protect the archaeological sites of Mexico? | 4 (2.4%) | 3 (1.8%) | 22 (13.2%) | 31 (18.6%) | 107 (64.1%) |
| Do you think that holographic devices could be used to help ensure the sustainability of the archaeological sites in Mexico? | 3 (1.8%) | 3 (1.8%) | 25 (15%) | 28 (16.8%) | 108 (64.7%) |

These technologies and their use for archaeological sites appear to be accepted, since 64.1% of the respondents were in favor of their use within archaeological sites to protect the sites, without negatively impacting the visitor experience at these sites. This could be a reflection of the previously observed outcomes, since responses were mostly positive. Based on participant opinions, projects with these technologies should be implemented for ensuring the sustainability of these sites, since they could support the dissemination of information about the archaeological sites without damaging or negatively impacting the environment.

According to studies carried out at the Manchester Autonomous University, technology has impacted the management and commercialization of places. In particular, in the context of cultural heritage, recent research explored the opportunity to integrate cutting-edge technologies such as VR and AR to improve the tourist experience [36].

Table 9 provides a comparison with Dieck et al. [26], who used an innovative virtual and immersive technology (AR) for studying the attraction of visitors to these types of technologies from several aspects.

Similar aspects were considered in both studies, confirming a positive acceptance of VITs by users. Regarding the importance of entertainment, educational experience and aesthetics are fundamental to studies focused on emerging technologies.

**Table 9.** Comparison with other studies using virtual and immersion yechnologies.

| | Historical-Cultural Sustainability Model for Archaeological Sites in Mexico Using Virtual Technologies | | Determining Visitor Engagement through Augmented Reality at Science Conferences: An Experience Economy Perspective |
|---|---|---|---|
| Participant general information | Average age of respondents: 34 years, more than half had college degree, followed by high school degree. The higher the educational level, the greater the degree of attraction. | Profile of Participants | Majority of respondents between 18 and 24 years, almost half had undergraduate degree, followed by postgraduate degree; more than half were students. |
| Knowledge of Virtual Technologies | Virtual and immersion technology most used by those surveyed who knew about virtual and immersion technologies was AR with 137 people | Aesthetics | Focused on the AR experience, attractiveness, and sense of harmony. |
| Educational Experience | Educational experience is positively impacted, since these technologies are viewed as tools for the diffusion of new learning. | Education | Focused on learning something new, knowledge, learning experience, and curiosity stimulation. |
| Entertainment | Positive impact on fun, demonstrating that the aforementioned technologies they increase people's entertainment experience. | Entertainment | Focused on the entertainment experience. |
| General Experience | Help the visitor to enjoy the experience of the place and who used such technology. | Escapism | Focused on using the AR as a way to scape reality |
| Aesthetics and Accessibility | Shows the opinion of the users, where the technologies are viewed as are simple to use, since, on many occasions, there are trained personnel who support their management, in addition to most mobile applications having an integrated tutorial or guide | Memories | Focused on memories made by AR |
| Sustainability | The acceptance of these technologies and their use at archaeological sites is positive; people are in favor of their use within archaeological sites to protect the heritage. | Satisfaction | General satisfaction with using AR technology |
| | | Visitor Engagement | Focused on the commitment to gain more knowledge about Manchester |

### 4.3. Comparison with Other Archaeological Sites in Mexico

The present work was provided to relevant authorities such as José Cuauhtli Alejandro Medina Romero, CEO of Xochicalco. Under Medina's leadership, Xochicalco has been one of the first archaeological sites where AR has been significantly used. When asked about the types of VIT used in tourism in Mexico, based on their experience with the archaeological site, he commented:

In Mexico, I have known very few cases where this type of technology has been used. However, my vision and knowledge are more focused on the field of in-house development such as the management of an Archaeological Site and an Archaeological Site Museum open to the public.

Based on his experience outside the area in his charge, he commented that he knew cases in which technologies such as AR and QR codes are used in other centers for tourist purposes, highlighting private museums that are directed by foundations. Hence, there had been no investment in this technology despite seeing the success in museums where they had already been applied.

Medina Romero commented on the application called Xochicalco AR, developed for the archaeological site, see Figures 10 and 11. In addition, he expressed knowing other archaeological sites where these technologies are used or were developed, e.g., Teotihuacán archaeological monument site, the El Teul archaeological site in Zacatecas, a prototype implemented in Tulum, as well as an application developed for the archaeological site of Templo Mayor in Mexico City.

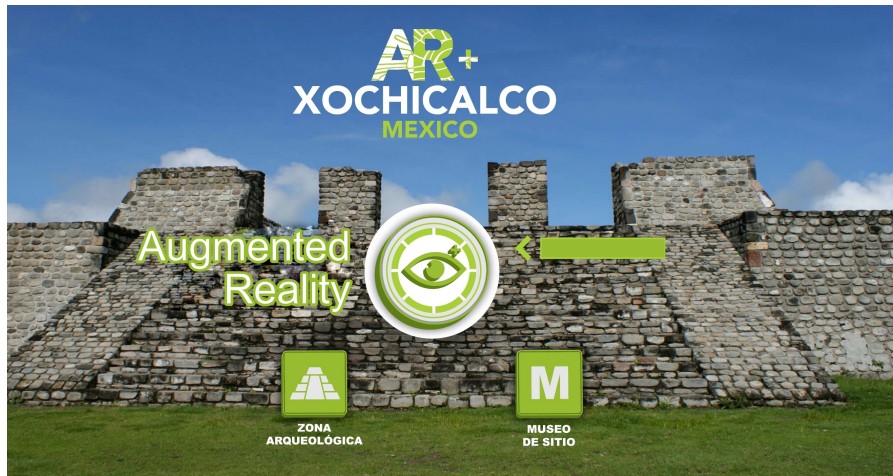

**Figure 10.** Xochicalco augmented reality (AR) application.

The managers of the tourist sites could make this type of technology accessible to their visitors, and Medina Romero described his experience as follows:

One of the advantages of implementing this type of technology is the dissemination of the knowledge of the cultural heritage of Xochicalco to visitors, which is why we integrate teams to develop, retake, and implement this type of technology at this World Heritage site, with a special goal to make them accessible to our visitors.

Medina Romero commented about the importance of the accessibility to other technologies such as the internet due as visitors would be unlikely to use AR without a connection to the internet. This underscores that infrastructure is required to adequately use any technology.

An important topic in the development of this research was how VITs support the dissemination and sustainability of the history and culture of archaeological sites. Per Medina Romero, these technologies have been effective and positive:

First of all, the information on Xochicalco throughout the entire site is offered in both in English and Spanish, so we are able to reach more visitors for this reason. The language in which they are produced is simple and understandable, so we can reach a good segment of visitors who speak Spanish.

In addition, he expressed that the level of acceptance of the diffusion with these holographic devices could be measured:

I would like to comment that the implementation of the auditory certificate allows monitoring the number of reproductions, which can be contrasted with the visit statistics and reflect positive relationships to develop these types of initiatives on the site. No doubt that the augmented reality application also reflects a positive effect on spreading the culture of this site.

In the particular case of Xochicalco, they have the conviction to continue developing digital educational tourism alternatives to ensure the sustainability of the cultural heritage of Xochicalco as a World Heritage Site, as we can see in the following words of Medina Romero:

The obtained results verified that it is an excellent tool for educational communication with a wide range of possibilities.

Regarding tourist guides, Medina Romero believes that, at this time, tourist guides do not have the training to use or rely on this type of technology. However, he suggested that, if the global trend leads toward the use of these technologies, they would have to adapt.

Regarding the use of VITs as an alternative for tourism, Medina Romero mentioned a real necessity:

Not only for tourists but in general to facilitate the transmission of knowledge for all visitors to cultural spaces and to promote understanding of what you want to convey. I am convinced that it is an alternative that must be considered in all cultural, educational, and tourist spaces.

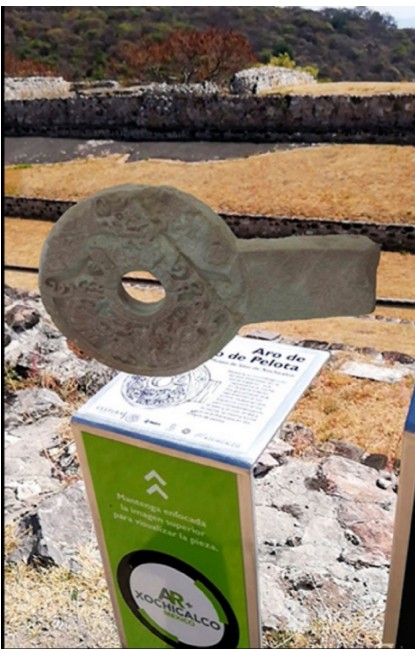

**Figure 11.** Xochicalco AR functioning within the archaeological site.

There is currently no plan for the sustainable conservation of national cultural heritage based on VITs if we refer to th established axis within the country's National Development Plan. However, Medina Romero commented that the Ministry of Culture contemplates this type of alternative in the national strategy to spread culture. With respect to the Xochicalco archaeological site, the management plan includes sustainability, education, dissemination, and research programs, considering constant development and eventually the implementation of new technologies for the digital preservation of this heritage worldwide and its eventual educational communication. As a final question, he was asked for his opinion on national tourism and if it is prepared for the use of VITs, to which he replied:

I have seen from the experience with the implementation since August 2017 and later in 2018 that, at the beginning, the use of these technologies is complicated or even a little slow; as time passes, a better coupling of visitors to the use of these technological tools is noted. I think it is a correct step that cultural sites should take and it could also be strongly promoted at tourist sites.

Based on interpretation of the above answers, our research is in line with the needs of creating a sustainable cultural-historical model for archaeological sites in Mexico due, to a large extent, to the use of VITs, which support the archaeological sites. VITs are well accepted by the tourists who visit these archaeological sites. The findings provide guidelines to start planning a conservation method using these technologies. As mentioned by Medina Romero, digital preservation could be relevant.

## 5. Conclusions

In this study, we selected the El Tepozteco archaeological site as a basis, starting from the problem of the damage caused by conventional tourism and the lack of its cultural and historical diffusion, as well as the negative impacts of tourism on the environment that is contrary to sustainability. The site selection was relevant since it allowed us to determine that virtual and immersion technologies are now an important component of reducing the impact caused by visitors to archaeological sites. Likewise, VITs improve the user experience, improving satisfaction. In structuring the theoretical-conceptual framework, we reviewed the literature on several concepts and theories for the development of this research, some of them related to VITs, helping enrich the finding. This literature review allowed us to identify the problem faced by various tourist sites.

The application of these technologies is still lacking in tourism in Mexico, since national studies about their use at tourist sites are lacking; we found that the use of these technologies is increasing

in popularity, so they are relatively new to tourists and researchers. During the diagnosis of the current situation of the system under study, we interpreted the existing advantages and disadvantages, in conjunction with collecting the opinions of the local population and a manager of the archaeological site to identify existing and non-existent relationships to more precisely understand the problems facing the system. The most notable advantage of VITs is the ability to generate new experiences that improve learning and entertain people, while preventing damage to the environment. However, there are disadvantages in its application such as the cost of implementation, the cost of content development, and the quality of audio and video, although with the correct management of resources and creating teams in well-designed workplaces, opportunities can be created.

From the outcomes yielded by the analysis, a sustainable historical-cultural model was constructed with VITs for an archaeological site in Mexico based on the systems thinking paradigm, using El Tepozteco as a case study. This model allowed us to establish the interrelationships of the agents to avoid conflicts and to generate better strategies.

An alternate prototype to the holographic pyramid was designed to meet the established objective to allow the dissemination of the history and/or culture of the El Tepozteco archaeological site. Despite our proposal not being directly applicable to the site under study, it was generated with the objective of satisfying the needs of future users. The current situation and the prototype were contrasted to validate our prototype. The information collected demonstrated the current situation of the system. In contrast, VITs have not yet been sufficiently developed or used at El Tepozteco due to several conflicts:

- Lack of use of VITs at the archaeological site.
- Mismanagement of the resources granted to the archaeological site.
- Lack of sustainability strategies implemented with the help of VITs.
- No historical-cultural diffusion of the archaeological site under study.
- Lack of training for tourist guides on new technologies.
- The damage generated by tourism to the site.
- Lack of legislation for the use of VITs at archaeological sites.

These conflicts present opportunities for inhabitants around the archaeological site and those in charge. The proposed prototype allowed a series of desired and feasible changes to be generated to ensure historical-cultural diffusion with the help of VITs at the archaeological sites occurs harmoniously with all its actors, including economic, environmental, social, and cultural sustainability, among other features. Notably, the feasibility of these changes depends on the resources and the population.

The interview with Medina Romero yielded important data about people's knowledge of VITs. After analyzing the results, we concluded that this type of technology positively impacts the user experience, which shows that AR, VR, and holography should be used to improve the perception of visitors to tourist sites.

In the present work, the main objective was fulfilled. The SSM was used to develop seven stages to generate a general diagnosis of the situation and construct a proposal of desirable and sustainable changes. This was the basis of demonstrating the importance of considering VITs as a method of disseminating sustainable history and culture. In the places where they have been implemented, the success has been visible. Hence, the hypothesis proposed in the research is supporting, corroborating that these technologies improve the tourist experience and contribute to the preservation of historical and cultural heritage.

The sustainability model developed in this work will allow archaeological sites to innovate in the dissemination of history and culture, producing a better experience for visitors and the local inhabitants, while considering the conservation of the environment. Therefore, our proposal will promote the sustainable use of VITs.

Based on the results of this research, new lines of research emerge. The SARS-CoV-2 COVID-19 pandemic has posed new challenges for tourism worldwide, so this type of technology can be implemented, providing an opportunity for the developed model to include remote experiences

when it is not possible to physically visit the tourist site, and to allow its operation within the new normal without neglecting sustainability.

The development of virtual and immersion technologies is advancing quickly, so it would be prudent to investigate how the aspects of accessibility and social inclusion can be added, allowing people with disabilities to make use of these technologies to help improve their tourist experience.

Regarding the literature review, the work performed at a national level on the impact of VITs in tourism is limited, so this work can be used as a basis for future studies on VITs in archaeological zones, as well as tourist destinations in Mexico, since there are many areas of opportunity for the development of research on this topic.

Finally, from the current situation, sustainable innovation can be achieved in tourism using virtual technologies, since a greater use of these technologies is required due to physical distancing requirements, so we recommend continuing with this sort of investigation to produce an application that benefits both tourism and the environment.

**Author Contributions:** Conceptualization, A.G.A.A.and J.J.M.E.; methodology, software and validation, J.J.M.E. formal analysis, R.T.P.; investigation and resources, O.M.M.; data acquisition, A.A.; writing original draft preparation, J.J.M.E., A.G.A.A., and O.M.M. All authors have read and agreed to the published version of the manuscript.

**Funding:** This study was supported by the National Polytechnic Institute (Instituto Poliécnico Nacional) of Mexico under project Nos. 20200638, 20200324, and 20202161, granted by the Secretariat of Research and Postgraduate(Secretería de Investigación y Posgrado), National Council of Science and Technology of Mexico (CONACyT).

**Acknowledgments:** The research described in this work was conducted at Escuela Superior de Turismo and Escuela Superior de Ingeniería Mecánica y Eléctrica of Instituto Politécnico Nacional, Campus Zacatenco. This research was part of the master's thesis entitled Modelo de Difusión Histórico-Cultural con Tecnologías Virtuales para una Zona Arqueológica de México basado en el Paradigma Sistémico supported by Geovanni Ambrosio and directed by Ricardo Tejeida and Jaime Moreno.

**Conflicts of Interest:** The authors declare no conflict of interest.

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
