# Peer review of "Historical-Cultural Sustainability Model for Archaeological Sites in Mexico Using Virtual Technologies"

_sustainability, doi:10.3390/su12187337_

Round 1

Reviewer 1 Report

This is a very interesting paper about the use of holography for the presentation of artifacts in archeological sites. Authors suggst the implementation of a holographic system for presenting information in archeological sites in order to protect them from misuse during visiting, as a part of a whole methodology that ensures sustainability.

The paper presents their idea, it includes related work and the presentation of the methodology. However, there are a lot of isues that authors are proposed to address in order to increase the quality of the paper.

Firstly, extensive editing is needed as there are various syntax errors that prevent seamless reading. Additionally, more documentation and linking is needed among the various parta of the paper.

Following are certain examples that show these issues.

206 The priority in heritage management is the conservation and restoration of this archaeological
207 site, the reason to the dissemination of its historical and cultural wealth to society is sometimes slowed.
It needs rephrasing. Dows not make sense as written.

211 Taking into account the aforementioned weaknesses that appear in other Archaeological Sites
212 of Mexico and that the competition of this type of tourist attractions is increasing [24], when making
213 a comparison it is required to strengthen this section within the El Tepozteco Archaeological Site, not
214 only to prioritize the management of this Tourist Attraction but also to increase the demand in a
215 responsible way and *too* generate income that can support restoration, conservation and support for
216 the continuous study of the area.
Very long sentence. Combined with an error (use of too), meaning is lost.

239 ...allowing all visitors to access to that technology without needing to bring a mobile
240 *holographic device* for
What kind of device would they bring in this solution? What is needed in the alternate solution?

234 As well as the Augmented Reality program managed in the *Archaeological Site of Xochicalco*,
235 and the system of auditory documentation linked by QR code that contribute to the conservation and
236 dissemination of the cultural heritage itself that has brought positive results in the visitor’s experience.
Sudden presentation of a solution for thw site of Xochicalco without adequate linking with the rest of the text

242 & 4.2 The *alternative* prototype to the Holographic Pyramid works by projecting ...
Main and alternate/alternative prototypes should be presented more clearly

272 ... Within the methodology, it was planned to use a Holography prototype to compare with
273 the conceptual model using the “Hologram Generation System” [25] as an alternative methodology.
The implementation process is shown adequately. The operation process should be presented as well.

Soft Systems Methodology steps
You present the methodology starting at step 4 (conecptual model). The first steps should be covered.

Figure 3 Conservation / Environment
It is not shown how the two parts assure conservation and the other two the protection of the environment.

capture, encoding, and rendering.
The same names should be used for the implementation steps everywhere.

327 The first
328 element is the Holographic Projection Device, projecting a video of the target with a black background
329 in the *Holographic Pyramid*.
What is the value of projecting a video in such device? Please explain. More details are needed about figure 6

Once again question categories of survey do not match the presented ones. There sould also be a differentiation between previous experience and hands-on experience. It should be made more clear how the use of main and alternativee prototypes are deifferentiated during survey.

The comparison should not cover only native museums but also international museums and sites with similar technology, such as the following:

Wang S., Osanlou A., Excell P.S. (2017) Case Study: Digital Holography as a Creative Medium to Display and Reinterpret Museum Artifacts, Applied to Chinese Porcelain Masterpieces. In: Research and Development in the Academy, Creative Industries and Applications. SpringerBriefs in Computer Science. Springer, Cham

Alejandro Madrid Sánchez, Leidy Marcela Giraldo, Daniel Velásquez Prieto, "Monocolor and color holography of pre-Hispanic Colombian goldwork: a way of Colombian heritage appropriation," Proc. SPIE 10558, Practical Holography XXXII: Displays, Materials, and Applications, 1055803 (19 February 2018); https://doi.org/10.1117/12.2291190

Reviewer 2 Report

Dear authors:

Thank you very much for your contribution to the protection of cultural heritage. The use of new technologies applied to cultural heritage has great potential today and is a very innovative field of study. I encourage you to continue working on this topic.

I have read your article carefully, in my opinion, the weakness of this work lies in the fact that it is a very specific case study that does not fully contribute to the advancement of scientific knowledge on the topic. In order to solve this problem, I suggest the following recommendations

  1. Taking into account that the fundamental objective of this paper is the use of new technologies in the interpretation of cultural heritage, I miss a bibliographic review about national and international case studies related to this topic. There is a large bibliography about this topic, in the following paper you can take a look some of the most significant cases: The Historic City, Its Transmission and Perception via Augmented Reality and Virtual Reality and the Use of the Past as a Resource for the Present: A New Era for Urban Cultural Heritage and Tourism?

2. I miss the main and secondary objectives as well as the working hypotheses or research questions being clearly explained. In the same way, it must be clearly indicated in the conclusions if the research objectives are reached and if the working hypotheses are corroborated. Future lines of research should be pointed out in depth.

3. The survey is the strong point of the work. For a greater scientific relevance and an advance in knowledge of the topic I suggest that the results of the survey be contrasted and compared with other studies, national e international. In this way, it will be possible to observe how this case study is positioned with the existing scientific knowledge about the topic. 

I hope these suggestions help you to improve the work and can be published.    

Reviewer 3 Report

The main issue is grammar editing. You must improve the grammar or it will detract from the article as a whole. 

The topic is very significant and will be of high interest to multiple audiences. The illustrations and bibliography are good. It is overall well organized and clear. 

I think the Intro could be less general, and more exciting and engaging about the topic. Maybe use an example from a visitor/potential visitor perspective of how useful the technology would be. 

I recommend this for publication with the requirement that it is very well edited for grammar/punctuation/capitalization/etc. 

Round 2

Reviewer 1 Report

The authors took into account the comments. Some little effort is still necessary in order to ensure strong coherence among related work, the presentation of the method and the presentation of the results.

Reviewer 2 Report

Citations 30 and 31 should be reviewed in the body of the text.
